

# Development of a novel mathematical model that explains SARS-CoV-2 infection dynamics in Caco-2 cells

Vladimir Staroverov[1,2,*], Stepan Nersisyan[1,3,4,5,*], Alexei Galatenko[1,2], Dmitriy Alekseev[1,2], Sofya Lukashevich[1], Fedor Polyakov[1,6], Nikita Anisimov[1] and Alexander Tonevitsky[1,6]

[1] Faculty of Biology and Biotechnology, HSE University, Moscow, Russia
[2] Faculty of Mechanics and Mathematics, Lomonosov Moscow State University, Moscow, Russia
[3] Institute of Molecular Biology, The National Academy of Sciences of the Republic of Armenia, Yerevan, Armenia
[4] Armenian Bioinformatics Institute (ABI), Yerevan, Armenia
[5] Current Affiliation: Computational Medicine Center, Sidney Kimmel Medical College, Thomas Jefferson University, Philadelphia, PA, United States
[6] Shemyakin-Ovchinnikov Institute of Bioorganic Chemistry, Russian Academy of Sciences, Moscow, Russia
* These authors contributed equally to this work.

Corresponding author
Alexander Tonevitsky,
atonevitsky@hse.ru

## ABSTRACT

Mathematical modeling is widely used to study within-host viral dynamics. However, to the best of our knowledge, for the case of SARS-CoV-2 such analyses were mainly conducted with the use of viral load data and for the wild type (WT) variant of the virus. In addition, only few studies analyzed models for *in vitro* data, which are less noisy and more reproducible. In this work we collected multiple data types for SARS-CoV-2-infected Caco-2 cell lines, including infectious virus titers, measurements of intracellular viral RNA, cell viability data and percentage of infected cells for the WT and Delta variants. We showed that standard models cannot explain some key observations given the absence of cytopathic effect in human cell lines. We propose a novel mathematical model for *in vitro* SARS-CoV-2 dynamics, which included explicit modeling of intracellular events such as exhaustion of cellular resources required for virus production. The model also explicitly considers innate immune response. The proposed model accurately explained experimental data. Attenuated replication of the Delta variant in Caco-2 cells could be explained by our model on the basis of just two parameters: decreased cell entry rate and increased cytokine production rate.

## INTRODUCTION

Severe acute respiratory syndrome coronavirus 2 (SARS-CoV-2) replicates in different types of human cells causing Coronavirus Disease 2019 (COVID-19). The primary site of SARS-CoV-2 infection is the lungs, and the acute respiratory distress syndrome is one of the main causes of high mortality (*Gibson, Qin & Puah, 2020*). At the same time, the virus can actively replicate in the intestine (*Qian et al., 2021*), which can explain frequent

gastrointestinal symptoms of the disease. Moreover, several studies reported prolonged viral shedding in the intestinal cells, which is suggested to play an important role in long COVID-19 (*Natarajan et al., 2022*; *Zollner et al., 2022*). Of note, human colorectal cancer cell line Caco-2 became one of the most popular *in vitro* models for studying SARS-CoV-2 replication, since after differentiation these cells naturally express ACE2 (the main receptor for SARS-CoV-2) and have clear enterocytic phenotypes (*Knyazev, Nersisyan & Tonevitsky, 2021*).

Given experimental data (such as viral load measurements at several time points) it is challenging to recover the values of viral life cycle parameters. Within-host mathematical modeling is a popular technique for the inference of such parameters (*Ciupe & Heffernan, 2017*). Parameter inference could be beneficial for comparative analysis of different virus variants. In the context of COVID-19, the overwhelming majority of reports are based on the fitting of differential equation model coefficients to the RT-qPCR analysis data of patients' nasopharyngeal swabs. Such a technique allowed researchers to infer key parameters of viral kinetics in humans (*Hernandez-Vargas & Velasco-Hernandez, 2020*), describe the main differences between SARS-CoV-2, SARS-CoV and MERS-CoV replication dynamics (*Kim et al., 2021b*), study COVID-19 treatment strategies with antiviral drugs (*Ohashi et al., 2021*; *Iwanami et al., 2021*; *Kim et al., 2021c*), make an accurate estimations of the incubation period of COVID-19 (*Ejima et al., 2021*) and compare different criteria for ending isolation for COVID-19 patients (*Jeong et al., 2021*). The authors of another study found significant differences in viral life cycle parameters in young and aged macaques (nasal and throat swabs were analyzed separately) and applied their model to gain insights on antiviral treatment (*Rodriguez & Dobrovolny, 2021*). *Heitzman-Breen & Ciupe (2022)* combined within-host and aerosol mathematical models to study the interplay between upper respiratory viral kinetics and between-host virus transmission using data from SARS-CoV-2-infected hamsters.

However, there are multiple cell types which could be infected by SARS-CoV-2 and multiple forms of immune response (*e.g.*, innate immunity, T-cell immunity, antibody response). As a result, the inferred parameters get "contaminated". For example, death rate of an infected cell also indirectly includes the effect of cellular immunity (since killer T-cells eliminate virus-infected cells). Given that, fitting model parameters to *in vitro* infection data is a promising strategy for the more precise explanation of experimental data because of the higher cell homogeneity, the absence of adaptive immune response and innate immunity cells (limited immune response could originate from infected cells). To the best of our knowledge, only *Bernhauerová et al. (2021)* considered within-host mathematical models in the context of *in vitro* infection data. Ultimately, the authors showed significant differences in infection rates and incubation times when infecting A549-ACE2 (human lung epithelium cells) and Vero-E6 (green monkey kidney cells) cell lines with SARS-CoV-2. Within the parameter inference procedure the authors used only virus titers data (RT-qPCR of cell culture supernatant), and only one virus variant was considered.

To close the aforementioned gaps we collected multiple types of publicly available longitudinal experimental data derived from SARS-CoV-2-infected Caco-2 cell lines,

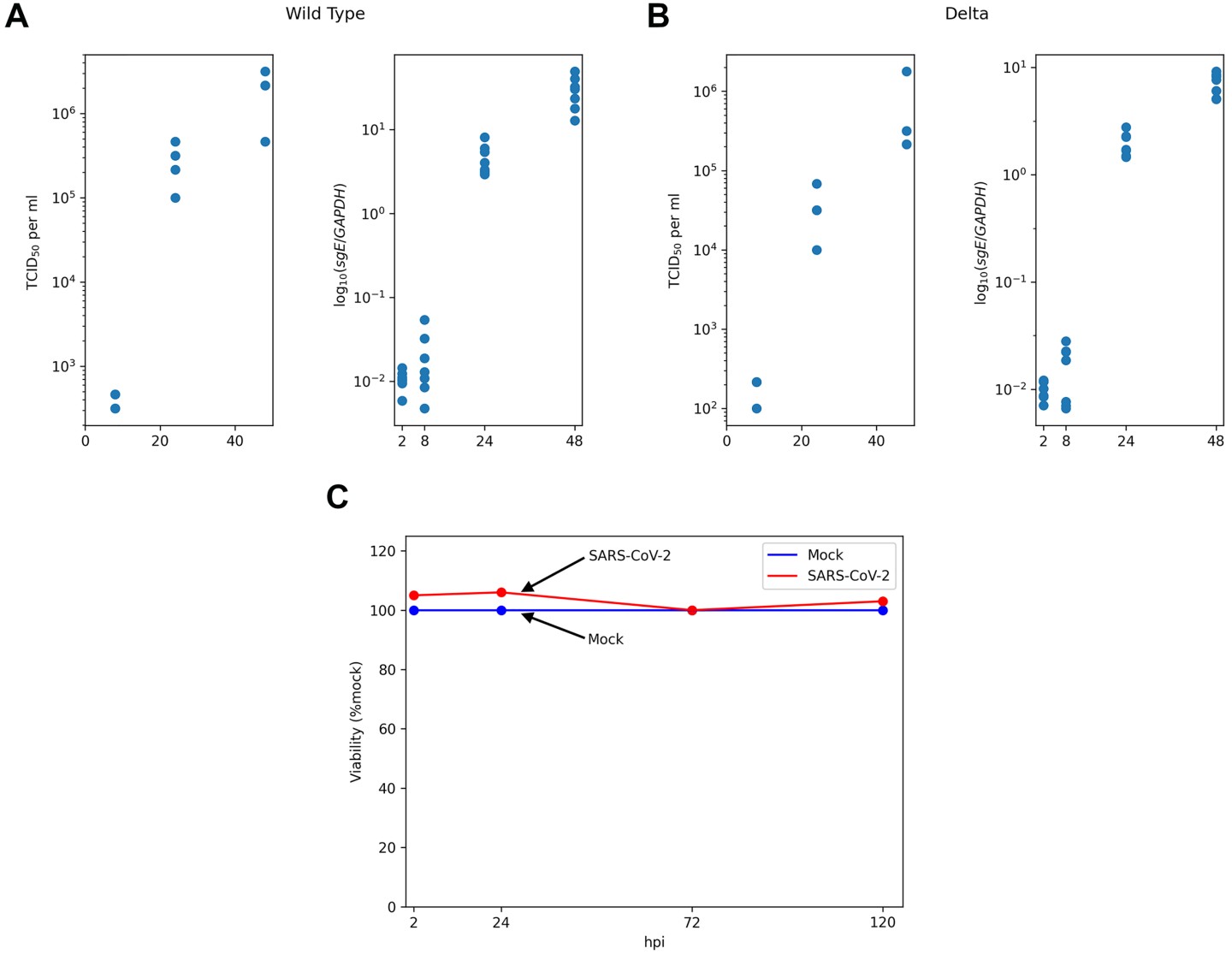

**Figure 1 Experimental data for SARS-CoV-2-infected Caco-2 cells.** TCID$_{50}$ assay and RT-qPCR data are shown for the WT (A) and the Delta (B) variants. Cell viability data is shown for the WT virus (C).

including measurements of infectious virus titers, intracellular viral RNA, number of infected cells and cells viability (Fig. 1). Based on a qualitative analysis of the experimental data we developed a novel mathematical model, which explicitly accounted for a number of intracellular viral RNA, cellular resources for viral particle production and the innate immune response. The model was validated by fitting its parameters to the data for the wild type (WT) and Delta variants of virus.

# MATERIALS AND METHODS

## Experimental data

The main experimental data were obtained from the recent work of Shuai with co-authors (*Shuai et al., 2022*). Specifically, human colorectal adenocarcinoma cell lines Caco-2
(30,000 cells) were incubated 2 h with SARS-CoV-2 at multiplicity of infection (moi) of 0.1 ($t = 0$ stands for the start of incubation). After incubation cells were washed and media was replaced. Next, SARS-CoV-2 titers were assessed by $TCID_{50}$ assay at three points: $t = 8, 24, 48$ h post infection (hpi). The number of subgenomic envelope viral RNA (*sgE*) relative to the human housekeeping gene *GAPDH* was measured by RT-qPCR at four time points ($t = 2, 8, 24, 48$ hpi). Thus, RT-qPCR data included only intracellular viral RNA abundance and did not take into account the genomic RNA in virus particles. The data was obtained from the "Source Data Fig. 1" and "Source Data Extended Data Fig. 1" files deposited alongside with the manuscript (*Shuai et al., 2022*). The results of cell viability assays ($t = 2, 24, 72, 120$ hpi) and the quantification of infected cells by N protein staining at $t = 24$ hpi under the same experimental conditions were obtained from two other papers of the same group: "Fig. 3A" in *Chu et al. (2020)* and "Fig. 2E" in *Shuai et al. (2020)*, respectively. The absence of cytopathic effect during SARS-CoV-2 infection in Caco-2 cells was also reported in multiple studies by other research groups (*Wurtz et al., 2021*; *Zupin et al., 2022*; *Alexander et al., 2020*; *Bartolomeo et al., 2022*; *Saccon et al., 2021*), including results for the Delta (*Mautner et al., 2022*) and the Omicron (*Dighe et al., 2022*) variants.

## Standard ODE models

Usually, differential equation models of within-host viral dynamics of SARS-CoV-2 take into account the number of SARS-CoV-2 viral particles ($V$) and the numbers of uninfected ($U$) and infected ($I$) cells at the time moment $t$ after infection (*Hernandez-Vargas & Velasco-Hernandez, 2020*; *Fadai et al., 2021*; *Du & Yuan, 2020*):

$$\begin{cases} \dfrac{dU}{dt} = -\beta UV \\ \dfrac{dI}{dt} = \beta UV - \delta I \\ \dfrac{dV}{dt} = pI - cV \end{cases} \tag{1}$$

Here uninfected cells become infected proportional to the number of uninfected cells and virus particles at rate $\beta$, infected cells produce new viral particles at rate $p$, infected cells and viral particles decay at *per capita* rates $\delta$ and $c$, respectively (the latter implicitly includes immune system response). At the initial moment of time a healthy host ($U(0) = U_0$, $I(0) = 0$) becomes infected with $V(0) = V_0$ viral particles. A similar model was used by *Bernhauerová et al. (2021)* to explain *in vitro* SARS-CoV-2 infection data. However, we showed that the solutions of these differential equations contradict to the cell viability data (see Results).

## The models proposed

We used two models of the SARS-CoV-2 infection process: the "discrete" model and the "continuous" model. The discrete model in essence implements the approach proposed by Doob et al. and Gillespie (see *e.g. Gillespie, 1977*), but is additionally endowed with a number of continuous parameters, *i.e.* the ones that take real values, so possibly a more correct term is "semi-discrete". The continuous model can be viewed as the limit of the
discrete model; it is the system of integro-differential equations in the variables $U_0$ ("regular" healthy cells), $U_1$ (resistant cells), $Cyt$ (cytokines), $I$ (infected cells), $V$ (free virions) and $R$ (intracellular viral RNA).

## Discrete model

At a high level the model consists of a discrete part and a continuous part. The discrete part is comprised by healthy, infected, resistant and exhausted cells. The continuous part describes the global state (the number of free virions $V$ and cytokines $Cyt$) and local states of infected cells (concentration of viral RNA denoted by $r$ and resources such as lipids denoted by $l$). The list of parameters of the proposed models is shown in Table 1 (parameter inference procedure is outlined below the model description).

The model operates in discrete time, *i.e.* at certain moments of time $t_1, t_2, \ldots, t_n, \ldots$ there occur events that modify either the discrete or the continuous part of the model. An event can be of one of the following types:

1. immunization of a healthy cell;

2. infection of a healthy cell;

3. infection of a resistant cell;

4. "infection" of an infected cell;

5. update of the system state.

There is also an additional event that consists in washing; it occurs at the moment $t = \Delta_{t\,clean}$. In terms of the model washing makes $V$ and $Cyt$ equal to zero. In the framework of the experiment considered there are no cytokines at the moment of washing, so only the number of virions is affected. Based on the experimental data used (TCID$_{50}$ assays) and previous reports (*Bernhauerová et al., 2021*), we assumed that viral particles do not decay in the time frame of the experiment (48 h).

Let us describe the discrete model in detail.

**Discrete part**. The cells are stored in an array of the length $C_{initial}$ (we assume that cells do not die, so the number of cells remains constant). The first $U_0$ elements correspond to "regular" healthy cells, the next $U_1$ elements correspond to resistant cells, then go $I^a$ "regular" infected cells, the final $I^e$ elements are $I^e$ "exhausted" infected cells, *i.e.* the ones that do hot have enough resources to produce virions. Initially $U_0 = C_{initial}$, $U_1 = I^a = I^e = 0$. All healthy cells and all resistant cells in the model are considered to be indistinguishable; every infected cell has a set of specific parameters described below.

**Continuous part**. The global state of the system is the pair $(V, Cyt)$, where $V$ is the number of free virions, $Cyt$ is the number of cytokines. Initially $V = V_{initial}, Cyt = 0$. Every infected cell has the following local parameters: $t^{inf}$ (time of infection), $r$ (the concentration of the viral RNA), $l$ (the volume of resources). At the moment of infection $t^{inf}$ is set equal to the current time, $r = 1, l = L_0$. Note that the values of parameters do not have to be integers, they can take real values.

**Selection of the next event**. Every "regular" event type (*i.e.* not including washing) is endowed with the parameter $\lambda_i$ specifying intensity of the flow of events of this type. Assume

**Table 1  Parameters of the models.**

| Parameter | Type | Meaning | Value and confidence interval (if applicable) | | |
|---|---|---|---|---|---|
| $\Delta_{t\,clean}$ | Fixed | Time to washing (h) | 2 | | |
| $L_0$ | Fixed | Initial resource concentration (unit) | 1 | | |
| $\Delta_{t\,cyt}$ | Fixed | Time from cell infection to cytokine generation (h) | 5 | | |
| $C_{initial}$ | Fixed | Initial number of healthy cells (pcs) | 30,000 | | |
| $V_{initial}$ | Fixed | Initial number of virions (pcs) | 3,000 | | |
| $N_V$ | Fixed | The number of points in time for free virion measurements (pcs) | 3 | | |
| $N_{RNA}$ | Fixed | The number of points in time for viral RNA measurements (pcs) | 4 | | |
| $\tau_1$ | Fixed | Time corresponding to the first measurement of infected cells (h) | 2 | | |
| $\gamma_{\tau_1}$ | Fixed | The fraction of infected cells for WT at the moment $\tau_1$ (%) | 6 | | |
| $\beta_{cyt}$ | Global | Intensity of cytokine entry $((pcs \cdot h)^{-1})$ | 1.18025e−08 [1.1241e−08 to 1.3663e−08] | | |
| $n_{l2V}$ | Global | Resources used to produce a single virion (fraction of the relative unit) | 0.000067 [0.00006−0.000073] | | |
| $p_{Vir}$ | Global | Virion generation intensity $((pcs \cdot h)^{-1})$ | 316,245 [80672.2−∞] | | |
| $\Delta_{t\,RNA\,double}$ | Global | Time required to double viral RNA in an infected cell (h) | 30 [25.9151−∞] | | |
| $k$ | Global | Reduction of infection intensity for infected and resistant cells (times) | 115 [99.3414−154.112] | | |
| $\Delta_{t\,latent}$ | Global | Time from cell infection to virus generation (h) | 7.36095 [7.36095−7.36095] | | |
| $Norm_V$ | Global | Normalization coefficient for the number of virions | 2.1 | | |
| $Norm_{RNA}$ | Global | Normalization coefficient for the concentration of viral RNA | 2.73 | | |
| $\beta$ | Local | Intensity of cell infection $((pcs \cdot h)^{-1})$ | WT 3.476e−08 [3.15e−08 to 3.65e−08] | | Delta 2.046e−08 [1.77e−08 to 2.15e−08] |
| $p_c$ | Local | Cytokine generation intensity $((pcs \cdot h)^{-1})$ | WT 106,422 [96,527−123,197] | | Delta 150,453 [136,465−201,622] |
| $R_0$ | Internal | Internal cell parameter updating frequency $(h^{-1})$ | 100 | | |
| $\Delta_{DE}$ | Internal | Step in numerical solution of the system of equations (h) | $10^{-3}$ | | |
| $\Delta_h$ | Internal | Step in evaluation of the function $p$ (h) | $10^{-5}$ | | |
| $\alpha_i, i = 1, \ldots N_V$ | Internal | Weights for penalties in the error functional | 1, 1, 1 | | |
| $\alpha'_j, j = 1, \ldots, N_{RNA}$ | Internal | Weights for penalties in the error functional | 10, 1, 1, 1 | | |
| $\theta_{\tau_1}$ | Internal | Weight for penalty in the error functional | $5.10^{-7}$ | | |

that current time equals $t$. For the case of immunization (type 1) $\lambda_1 = \beta_{cyt} U_0(t) Cyt(t)$; for the case of infection of a "regular" healthy cell (type 2) $\lambda_2 = \beta U_0(t) V(t)$; for the case of infection of a resistant cell (type 3) $\lambda_3 = \frac{\beta}{k} U_1(t) V(t)$; for the case of "infection" of an infected cell (type 4) $\lambda_4 = \frac{\beta}{k}(I^a(t) + I^e(t)) V(t)$; finally for the case of parameter updating (type 5) $\lambda_5 = R_0$. The time of the next "regular" event $t + \Delta_t$ is specified by the equality

$\Delta_t = 1/\sum_{j=1}^{5} \lambda_j$. In essence this equality is the expectation of an exponentially distributed random variable corresponding to the union of all five event flows. The type of the event at the moment $t + \Delta_t$ is selected randomly; it is equal to $i$ with the probability $\lambda_i/\sum_{j=1}^{5} \lambda_j$.

A detailed description of the implementation of events is presented in  Appendix S1.

### Continuous model

The continuous model can be viewed as the limit of the discrete model described above. The model is the system of integro-differential equations in the variables $U_0$ ("regular" healthy cells), $U_1$ (resistant cells), $Cyt$ (cytokines), $I$ (infected cells), $V$ (free virions) and $R$ (intracellular viral RNA):

$$U_0(t)' = -\beta U_0(t)V(t) - \beta_{cyt}Cyt(t)U_0(t)$$

$$U_1(t)' = -\frac{\beta}{k}U_1(t)V(t) + \beta_{cyt}Cyt(t)U_0(t)$$

$$Cyt(t)' = p_c I(t - \Delta_{t\,cyt}) - \beta_{cyt}Cyt(t)U_0(t)$$

$$I(t)' = \beta(U_0(t) + U_1(t)/k)V(t)$$

$$V(t)' = \int_0^t (\beta(U_0(t - x) + (U_1(t - x))/k)V(t - x))p(x)dx-$$

$$-\frac{\beta}{k}(I(t) + U_1(t))V(t) - \beta U_0(t)V(t)$$

$$R(t) = \int_0^t (\beta(U_0(t - x) + U_1(t - x)/k)V(t - x))\sigma(x)dx$$

Here $p(t)$ is the first derivative of the number of virions produced by an infected cell (evaluation of this function is described in Appendix S2), $\sigma(t)$ is the total concentration of viral RNA produced by a cell up to the time $t$ after infection ($\sigma(t) = 2^{t/\Delta_{t\,RNA\,double}}$). The first equation means that "regular" healthy cells become infected with intensity proportional to the product of the number of healthy cells and the number of free virions or become resistant with intensity proportional to the product of the number of healthy cells and the number of cytokines. The second equation means that resistant cells are produced from "regular" healthy cells and can be infected with intensity proportional to the product of the number of resistant cells and the number of free virions (note that the coefficient in the second equation differs from the coefficient in the first equation; in terms of Fig. 2 this coefficient is $\beta' = \beta/k$). The third equation is devoted to cytokines: new cytokines are produced by infected cells with a certain delay, and existing cytokines are used to make healthy cells resistant. The fourth equation describes infected cells which can be produced either from "regular" healthy cells or from resistant cells. The fifth equation is devoted to release of free virions. The first summand in essence means that infected cells produce free virions with time-dependent intensity $p(\tau)$ (note that the first factor under the integral sign is the right-hand side of the fourth equation). Speaking in more detail, this summand can be rewritten in the form
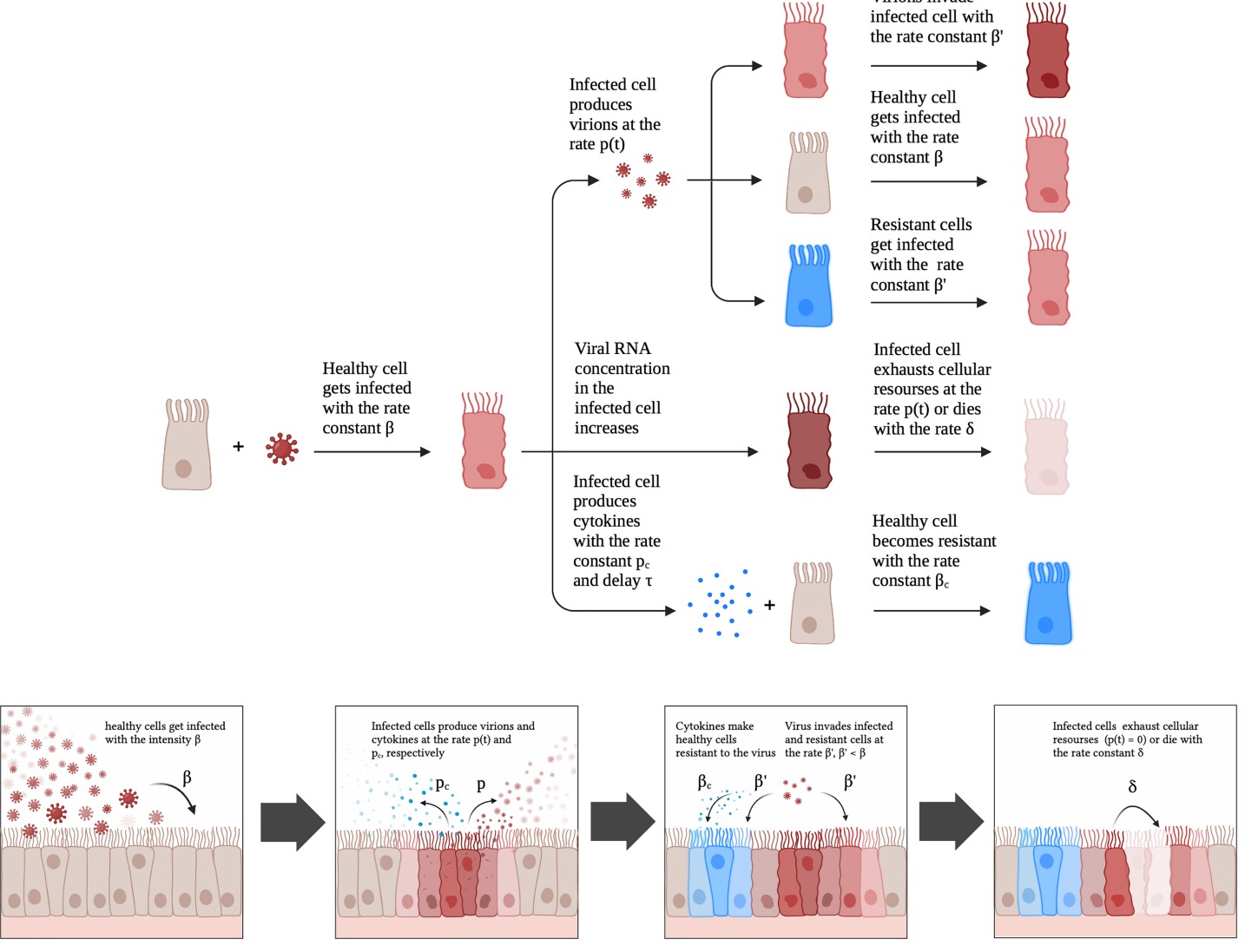

**Figure 2 A schematic illustration of the proposed model for SARS-CoV-2 infection dynamics in Caco-2 cells.**

$$\int_0^t (\beta(U_0(t-x) + (U_1(t-x))/k)V(t-x))p(x)dx = \int_0^t I'(t-x)p(x)dx$$

free virions are produced by cells infected at different time points. Every cell infected in the interval $[t-x, t-x+dt]$ (the number of such cells is $I'(t-x)dt$) at the interval $[t, t+dt]$ emits $p(x)dt$ virus particles. Thus the total number of virions produced by cells infected in the interval $[t-x, t-x+dt]$ is $I'(t-x)dt\, p(x)\, dt$. Hence, it only remains to integrate these values. The second summand shows that free virions are used to infect healthy cells but also may enter cells that are already infected. Finally the sixth equation computes concentration of viral RNA inside infected cells similarly to production of free virions in the fifth equation. The sixth equation was added to the system solely for the purpose of fitting to the existing data, as the variable $R(t)$ is not used in other equations.

Note that the sum of the first, the second and the fourth equation has zero in the right-hand side, thus the total number of cells is constant, *i.e.* cells do not die.

Computational approach for solving the proposed system of integro-differential equations is presented in Appendix S2.

## Selection of parameters of the proposed models

Our models operate with four types of parameters. The values of parameters of the type "Fixed" are extracted from literature or are directly inferred from data. The values of parameters of the type "Global" are assumed to be equal between the models corresponding to different virus types and are tuned using the whole dataset. One part of the "Global" parameters ($\beta_{cyt}$, $k$) is exclusively related to the properties of the target cells. The other "Global" parameters ($n_{l2V}$, $p_{Vir}$, $\Delta_{t\,RNA\,double}$, $\Delta_{t\,latent}$) are shared by the WT and the Delta viruses because we assume that the corresponding biological values could not be significantly altered by such a low number of mutations in viral genomic RNA. The values of parameters of the type "Local" are tuned using data corresponding to a single virus type. Finally, parameters corresponding to the implementation of the models are of the type "Internal".

### Selection of fixed parameters

The major part of fixed parameters were extracted directly from the experimental protocol: $\Delta_{t\,clean}$, $C_{initial}$, $V_{initial}$, $N_V$, $N_{RNA}$, $\tau_1$, $\gamma_{\tau_1}$. Time from cell infection to cytokine generation $\Delta_{t\,cyt}$ was set to 5 h based on existing literature for SARS-CoV-2 (*Neufeldt et al., 2022*; *Liu et al., 2022*; *Kim et al., 2021a*; *Stanifer et al., 2020*).

Due to parameter dependence it makes sense to make the parameter $L_0$ fixed and set it equal, *e.g.*, to 1. See "Parameter dependencies" for more detail.

### Selection of global and local parameters

Selection of global and local parameters was based on optimizing goodness of fit with respect to the following error functional.

Assume that *Params* is some tuple of parameter values. Then the error associated with *Params* is defined by the equality

$$Err(Params) = \sum_{i=1}^{N_V} \alpha_i (V_i - Norm_V \cdot v_i)^2 +$$

$$+ \sum_{j=1}^{N_{RNA}} \alpha'_j (R_j - Norm_{RNA} \cdot r_j)^2 + \theta_{\tau_1}(I(\tau_1) - C_{initial}\gamma_{\tau_1})^2$$

In this formula $v_i$ is the average of $\log_{10}$ of the number of virions in the experiment at the $i$th measurement, $V_i$ is $\log_{10}$ of the number of free virions in the model at the corresponding point of time, $r_j$ is the average of $\log_{10}$ of the concentration of viral RNA in the experiments at the $j$th measurement, $R_j$ is $\log_{10}$ of the value of viral RNA concentration in the model at the corresponding point of time, $I(\tau_1)$ is the numbers of infected cells in the model at the corresponding moment.

Note that *Err* accumulates all types of available data, namely the number of virions, concentration of viral RNA and the fraction of cells infected at the moment $t = 24$ hpi.

Functional optimization was performed with the help of coordinate descent method described in detail in Appendix S3. Global parameters were selected using the whole dataset, local parameters for concrete virus types were tuned using only data for the corresponding virus type.

Error functional *Err* was also used to obtain confidence intervals. For each parameter of interest the value was changed (decreased to get the left end, increased to get the right end) until the value of *Err* became 10% greater than the minimum value.

### Selection of internal parameters

The parameters $R_0, \Delta_{DE}, \Delta_h$ were selected in the following way. We started from the initial value 1 and performed parameter modification step (replace $R_0$ with $4R_0$, replace $\Delta_{DE}$ with $\Delta_{DE}/4$, replace $\Delta_h$ with $\Delta_h/4$) until the solution with modified parameters became indistinguishable from the solution for the original parameter values.

Weight parameters $\theta_{\tau_1}, \alpha_i, \alpha'_j$ were selected taking into consideration visual estimation of dispersion of the corresponding summands in the error functional *Err*.

### Parameter dependencies

Note that some of the model parameters are dependent. The pair $(L_0, n_{l2V})$ is such that modification of one of the pair elements can be compensated by modification of the remaining element so that model behavior does not change. Without loss of generality we set $L_0 = 1$ (*i.e.*, we modeled the fraction of cellular resources).

Pairwise independence of the remaining parameters was verified using Assertion 3 of Appendix S5. We considered all pairs of parameters and showed that minimum value of the functional generated by modifying respective parameters on some circle around minimum point of *Err* is essentially greater than the value in the center of the circle (see Appendix, Table S1). Assertion three implies that there is no pairwise dependence outside the circle considered.

# RESULTS

## Data description and motivation

We started with the qualitative analysis of previously published dataset of Caco-2 cells infected with the Wuhan (WT) and the Delta variants of SARS-CoV-2. Specifically, the measurements of infectious viral particles in medium, viral subgenomic RNA in infected cells and cell viability data were available for several time points after infection (Fig. 1). We also used the data on the percentage of infected cells 24 h post infection (see Materials and Methods for the details). Based on these data we made three key observations.

1. Caco-2 cells (including infected ones) stay viable throughout the whole experiment.
2. Both virus titers and intracellular viral RNA growth significantly slow down over time: 5 folds difference in the WT virus titers growth rates between 8–24 and 24–48 hpi intervals, 4.7 folds in the case of the WT viral RNA growth, 2.9 folds for the Delta virus titers, and 5.7 folds for the Delta viral RNA.

3. The fraction of infected cells after 24 h post infection is relatively small (approximately 6%).

Since the number of uninfected cells is sufficiently large throughout the whole experiment (point 3), exhaustion of healthy susceptible cells cannot explain the decrease in infection rate (point 2). This observation along with the absence of cytopathicity (point 1) led us to the hypothesis on the exhaustion of some key cellular resources needed for virus particle assembly and release (*e.g.*, lipids). We also hypothesized that the primary reason for the lack of infection in the majority of cells is connected to cell immunization (innate immune response).

The existing standard mathematical models (see Materials and Methods) were not applicable for explaining such data for several reasons. First, these models operate with the numbers of cells (uninfected, infected) and viral particles. Thus, measurements of intracellular viral RNA could be used for parameter inference only if we assume proportionality of viral RNA to the number of infected cells (which, as we believe, is a too strong assumption). Alternatively, one can infer parameters using only infectious virus titers data which ultimately results in unidentifiability of some parameters (*Bernhauerová et al., 2021*). Second, the absence of a cytopathic effect results in solution properties which contradict experimental data. Namely, in the case of the model with three variables ($U$, $I$, $V$) all cells become infected, which implies an unbounded increase in the number of viral particles, since they are produced by infected cells at a constant rate (see S4 Appendix for an accurate proof).

## A new mathematical model of SARS-CoV-2 infection *in vitro*

Given the aforementioned observations, we developed a mathematical model which included a possibility of cell exhaustion, direct modeling of intracellular viral RNA and the innate immune response. The model is schematically illustrated in Fig. 2. A healthy cell gets infected with rate constant $\beta$ upon contact with a productive virion. An infected cell produces virions after latent period $\Delta_{t\,latent}$ proportional to the number of viral genomes and cellular resources. Once a cell becomes exhausted (*i.e.*, resources for virion assembly/release run out), it can no longer produce new viral particles. The described scheme was initially explicitly implemented in the discrete model, while for the continuous differential equation model the virus production rate was modeled by the associated function $p(t)$ (Appendix, Fig. S1; see Materials and Methods for details). An infected cell also produces cytokines with the rate constant $p_c$ with a delay $\Delta_{t\,cyt} = \tau$. Cytokines then enter healthy cells with rate constant $\beta_c$ and make them more resistant to the virus; infection rate constant for resistant cells is $\beta' < \beta$.

## Parameter fitting and comparative analysis

For both WT and Delta viruses we optimized parameters of the mathematical model so that solutions fit the experimental data (virus titers, subgenomic RNA, percentage of the infected cells at $t = 24$ hpi). To simplify the computational procedure and improve interpretability, we assumed that most of the model parameters were the same for two

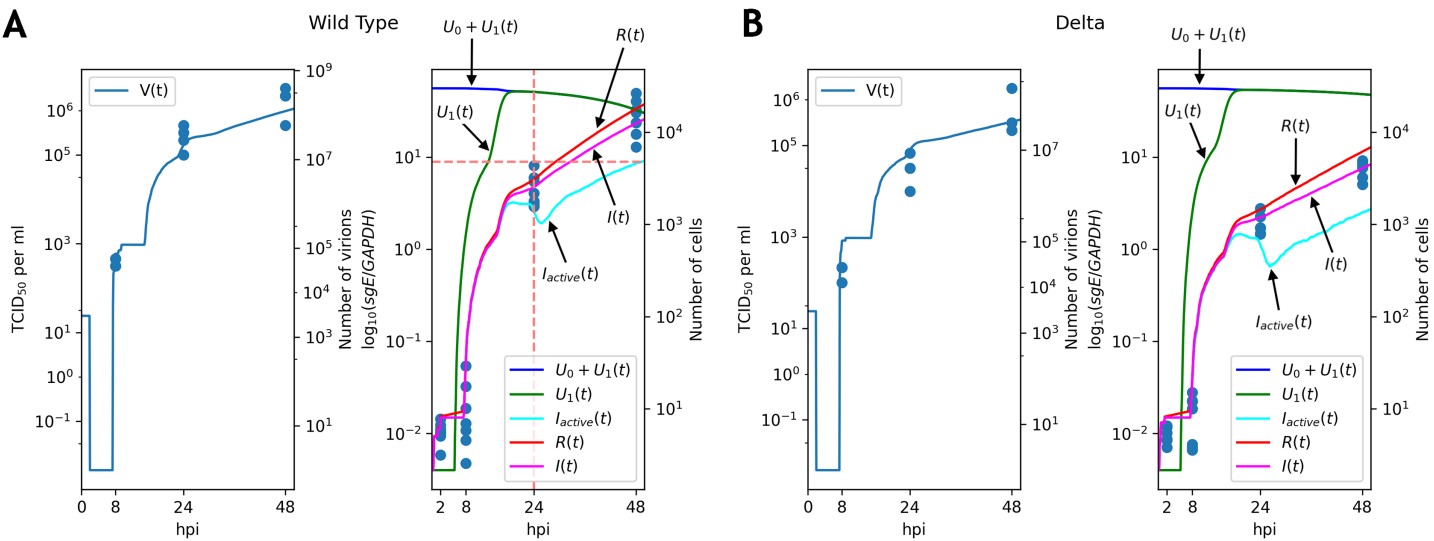

**Figure 3 Solutions of the discrete model accurately explain experimental data for the WT (A) and Delta (B) variants.** $TCID_{50}$ assay and RT-qPCR data are shown for the WT (A) and the Delta (B) variants. Cell viability data is shown for the WT virus.

SARS-CoV-2 variants. The exceptions were virus cell entry rate $\beta$ and cytokine production intensity $p_c$. For both variants the solutions explained the data well (Fig. 3; Appendix, Fig. S2). In both cases only a fraction of infected cells were active (*i.e.*, producing new viral particles) at $t = 48$ hpi. Notably, the discrete model allowed us to explicitly quantify the number of active infected cells (Fig. 3), which was not a case in differential equation model. Solutions for the remaining variables were near-identical between both models (Appendix, Fig. S2).

Note that graphs for the number of virions contain flat sections. The reason for the first flat section is the latent period (approximately 7 h, see the parameter $\Delta_{t\,latent}$ in Table 1): all virus particles are removed by washing, and new particles have not been emitted yet. The following flat segments are the reflections of the first one, since virus production period of an infected cell is short due to resource exhaustion (in terms of the continuous model the function $p(x)$ in the fifth equation of the system becomes equal to 0; see Appendix, Fig. S1), and new infected cells are in the latent period.

The key difference between replication profiles of two viruses was a significant reduction in the number of intracellular viral subgenomic RNA in the Delta-infected cells compared to the WT: 2.41 folds at 24 hpi, 4.2 folds at 48 hpi (Mann–Whitney's $U$-test $p = 2.16 \times 10^{-3}$ for both time points). A similar trend was also observed for infectious virus titers data, though the difference at 24 hpi (7.8 folds, $p = 0.029$) was higher than at 48 hpi (3.1 folds, $p = 0.057$). Our model captured these data well, extrapolating the same trends to the absolute (non-observable) numbers of viral particles and infected cells (Fig. 3). Both cell entry and cytokine production rates had different values as a result of parameter fitting to the WT and Delta data. Specifically, the rate of virus entry into Caco-2 cells was 1.7-fold higher for the WT compared to the Delta variant. Conversely, cytokine
_______________

production rate by an infected cell was higher for the Delta variant (1.4-fold). Confidence intervals for both parameters were not overlapping.

Full list of parameter values is presented in Table 1. We also verified that if all parameters except one are equal for WT and Delta, then approximation quality essentially drops (the least possible value of _Err_ in this case is 1.8 times greater than for the case considered above).

## DISCUSSION

We developed a novel mathematical model to explain SARS-CoV-2 infection dynamics _in vitro_. The key feature of the proposed framework was a low-level modeling of cell-virus interplay, including modeling of concentration of intracellular viral RNA and other abstract cellular resources needed for viral particle assembly (such as lipids). Innate immune response to the infection was also considered. The practical implementation of the model included both discrete modeling and differential equations. Discrete modeling allowed us to explicitly quantify some unobserved variables (_e.g._, the number of active infected cells), which was not possible with differential equations. At the same time, two models were essentially equivalent from the practical point of view, since they yielded very similar solutions for the common variables. We validated the proposed approach by simultaneously fitting model parameters to different types of data: infectios virus titers (TCID$_{50}$ assay), amount of intracellular viral subgenomic RNA (RT-qPCR), cell viability and percentage of infected cells (N protein staining). The resulting solutions accurately explained the data for both WT and Delta variants.

According to the inferred model parameters, attenuated replication of the Delta variant was due to the slower cell entry and the higher rate of innate immune response activation. Interestingly, these observations fit well with the clinical data. Namely, while COVID-19 caused by the Delta variant is more severe compared to the WT virus, gastrointestinal symptoms were significantly less frequent in the Delta case (_Hu et al., 2021_; _Fernández-de Las-Peñas et al., 2022_; _Schulze & Bayer, 2022_). Frequency of long COVID-19 was also significantly lower in the case of the Delta variant: odds ratio = 1.27, Fisher's exact test $p = 1.10 \times 10^{-6}$ (data for the WT and Delta long COVID-19 taken from _Sudre et al. (2021)_ and _Antonelli et al. (2022)_, respectively). Given the correlation of long COVID-19 and prolonged viral shedding in the intestinal cells (_Natarajan et al., 2022_; _Zollner et al., 2022_), our results are in line with the reduced long COVID-19 incidence for the Delta variant-infected individuals.

The main limitation of our analysis was the sparsity of experimental data in terms of the number of time points. Namely, system dynamics were not observable for large time intervals (_e.g._, 0–8, 8–24, 24–48 hpi), which indeed resulted in uncertainty of some model parameters like latent period $\Delta_{t\,latent}$. We also note that TCID$_{50}$ assay and RT-qPCR data are relative in their nature, so the model analysis required additional efforts to combine the absolute and the relative scales. Despite the mentioned limitations, the model allowed us to fit the experimental data on the qualitative level, which is unfeasible for the standard models.

The main direction of future research is application of the model to the same types of data collected for different cell lines and SARS-CoV-2 variants. Interestingly, there are several qualitative differences in viral replication between such setups. For example, all known variants of SARS-CoV-2 cause a strong cytopathic effect on African green monkey kidney VeroE6 cells, which is not the case for human cell lines (*Shuai et al., 2022*; *Chu et al., 2020*). Another example is a dramatically attenuated replication of the Omicron variant in a number of cell lines and animal models (*Shuai et al., 2022*; *Mautner et al., 2022*; *Suzuki et al., 2022*; *Halfmann et al., 2022*). The authors of several experimental reports explained such an alleviated Omicron replication by a loss of TMPRSS2 usage ability (*i.e.*, less efficient cell entry) (*Shuai et al., 2022*; *Meng et al., 2022*) and reduced interferon antagonism of the virus (*Bojkova et al., 2022a*, *2022b*).

## ACKNOWLEDGEMENTS

The authors thank Dr. Hin Chu for valuable comments and discussions.

### Funding

The research was performed within the framework of the Basic Research Program at HSE University. The funders had no role in study design, data collection and analysis, decision to publish, or preparation of the manuscript.

### Grant Disclosures

The following grant information was disclosed by the authors:
Basic Research Program at HSE University.

### Competing Interests

Stepan Nersisyan is an employee of Armenian Bioinformatics Institute (ABI).

### Author Contributions

- Vladimir Staroverov performed the experiments, analyzed the data, prepared figures and/or tables, authored or reviewed drafts of the article, and approved the final draft.
- Stepan Nersisyan conceived and designed the experiments, analyzed the data, authored or reviewed drafts of the article, and approved the final draft.
- Alexei Galatenko conceived and designed the experiments, analyzed the data, authored or reviewed drafts of the article, and approved the final draft.
- Dmitriy Alekseev performed the experiments, analyzed the data, prepared figures and/or tables, authored or reviewed drafts of the article, and approved the final draft.
- Sofya Lukashevich analyzed the data, prepared figures and/or tables, authored or reviewed drafts of the article, and approved the final draft.
- Fedor Polyakov analyzed the data, authored or reviewed drafts of the article, and approved the final draft.
- Nikita Anisimov conceived and designed the experiments, analyzed the data, authored or reviewed drafts of the article, and approved the final draft.

- Alexander Tonevitsky conceived and designed the experiments, analyzed the data, authored or reviewed drafts of the article, and approved the final draft.

## Data Availability

The full model description and parameter values are available in the Supplemental File.

## Supplemental Information

Supplemental information for this article can be found online at http://dx.doi.org/10.7717/peerj.14828#supplemental-information.

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
