# Peer review of "Development of a novel mathematical model that explains SARS-CoV-2 infection dynamics in Caco-2 cells"

_PeerJ, doi:10.7717/peerj.14828_

## Round 0.1 · original submission · Major Revisions

Please address the concerns raised by both reviewers and amend your manuscript accordingly.

Reviewer 1 ·

Basic reporting

The manuscript is clearly written.

Experimental design

I have some concerns with the experimental design here.
1. Parameter estimates are consistently presented without units. Numbers without units are meaningless.
2. There are no error estimates for any of the parameter values. There is also no assessment of identifiability of the parameters. This last point is interesting since the authors mention lack of parameter identifiability of previous parameter estimates as one of the motivations for this work.
3. The authors assume that wild-type and delta experiments differ in two particular parameters, but then conclude that differences in these two parameters are sufficient to explain differences in the observed experimental measurements. Perhaps other parameter pairs would also work; has this been checked? Also, without any error estimates, are the differences between the values really significant?
4. The authors seem to favour the discrete model fits, but to my eye, there isn't much difference between those fits and the continuous model fits. Since there is no assessment of the goodness of fit of either of these models, I can't judge whether one model actually explains the data better than the other.
5. For that matter, it's not clear that simpler models won't work. I'm not convinced by points 1-3 in section 3.1. The viral time courses (both intra- and extra-cellular) have only three time points, with both quantities largely showing the same trend. I'm not sure how the authors decided that one slows down growth at 24 hours while the other doesn't.

Validity of the findings

The claims that the innate response and cell exhaustion are required to fit this data as well as the claim that the infection rate and cytokine production rate are the only two differences between the delta variant and the original virus are not well supported by the evidence presented in the manuscript.

Reviewer 2 ·

Basic reporting

The article “Development of a novel mathematical model that explains SARS-CoV-2 infection dynamics in Caco-2 cells” addresses the temporal dynamics of SARS-CoV-2 infection in interferon-competent human colorectal adenocarcinoma cell line Caco-2 with mathematical modelling.
The article is written in clear English. The main objective of the paper is to explain experimental data obtained from in vitro infection with SARS-CoV-2 on the Caco-2 cells which show that SARS-CoV-2 infected cells seem to remain viable (up to 5 days) well beyond the termination of the infection experiments (up to 3 days). For this reason, the mathematical model, which accounts for virus-induced cell death cannot accurately describe virus infection in Caco-2 cells. The authors introduce a mathematical model, which considers the action of innate immunity as an explanatory variable responsible for the survival of infected cells. They fit their model to experimental data to calibrate the model.

The authors reference only 35 articles, the majority of which are experimental. Given the massive amount of modelling work that has been done on SARS-CoV-2 in humans (e.g., [1, 2]) and in vivo (e.g., [3]), it would be good to mention other papers and their relation to this work.

[1] E. A. Hernandez-Vargas, J. X. Velasco-Hernandez (2020) In-host Mathematical Modelling of COVID-19 in Humans. Annual Reviews in Control (50). https://doi.org/10.1016/j.arcontrol.2020.09.006.
[2] Heitzman-Breen N and Ciupe SM (2022) Modeling within-host and aerosol dynamics of SARS-CoV-2: The relationship with infectiousness. PLOS Comp Biol 18(8): e1009997. https://doi.org/10.1371/journal.pcbi.1009997
[3] Rodriguez T. and Dobrovolny H. M. (2021) Estimation of viral kinetics model parameters in young and aged SARS-CoV-2 infected macaques. R. Soc. Open Sci. 8202345202345. https://doi.org/10.1098/rsos.202345

Raw data used should be made available as a supplementary file and description of how they were extracted from other studies should be mentioned in Methods.

Experimental design

- In Introduction, line 67 – please, rephrase, as the data were obtained elsewhere and not generated in-house for this paper.
- Lines 83-84 – how were the data on infected cells incorporated in the model? I also couldn’t find the values or figures.
- Is the model (1) used at any point in the study? If not, it is redundant and should be removed.
- Lines 91-92 – The term (-delta*I) in the model (1) gives the mean duration 1/delta for which infected cells produce virus. The lack of viral particle production doesn’t necessarily mean that these cells are dead. There is evidence of upregulation of anti-apoptotic genes and down-regulation of pro-apoptotic genes in Caco-2 cells during SARS-CoV-2 infection [4]. On the other hand, cytopathic effects have been recorded in Caco-2 infected with SARS-CoV-2 [5]. I think that the term for infected cell removal needs to be accounted for.
- Figure 1 – There should be clearly stated where each dataset comes from and what the measurements mean, especially for genome copies.
- Lines 92-97 – How does this model reduction relates to the paper? It is used at any point or do authors use it for something? If not, it should be removed.
- The main models used to generate dynamics in Figures 3 and S2 are the models in the supplementary material in Appendix S1 and Appendix S2. They should be in the main text. Also, why not use only the continuous model?
- Why is the dynamics between 8 and 24h flat for some time in Figures 3 and S2 (and then jumps up)?
- Which methods did the authors used to fit models to data? It needs to be described.
- Lines 179-183 – I don’t agree. TCID50 increases by one log from 5 to 6, but the log RNA copy number from 3.5 to 4. Also, what exactly sgE/GAPDH refers to? How the two y-axis in the Figures 3 and S2 relate to one another?
- The values of parameters should be added to Table 1. The Table S1 is duplicated.
- Why didn’t the authors also analyse data from infection of Calu3 cells?
- Do Caco-2 cells also exhibit high viability when infected with Delta variant?
- Usually, point estimates are accompanied with parameter confidence intervals, which are missing in this study. Is there any reason for that? I’d suggest the authors provide them.
- I don’t see anywhere in the model equations that the infectious virus decays. Infectivity of virus is time-limited, but the only way it’s being removed is if it’s taken up by cells. I think it shouldn’t be omitted.

[4] Diemer, C., Schneider, M., Schätzl, H.M., Gilch, S. (2010). Modulation of Host Cell Death by SARS Coronavirus Proteins. In: Lal, S. (eds) Molecular Biology of the SARS-Coronavirus. Springer, Berlin, Heidelberg. https://doi.org/10.1007/978-3-642-03683-5_14
[5] Bojkova, D., Klann, K., Koch, B. et al. Proteomics of SARS-CoV-2-infected host cells reveals therapy targets. Nature 583, 469–472 (2020). https://doi.org/10.1038/s41586-020-2332-7

Validity of the findings

no comment

---

## Round 0.2 · Minor Revisions

Please address the remaining issues pointed out by reviewer 2 and amend the manuscript accordingly.

Reviewer 1 ·

Basic reporting

No comment

Experimental design

No comment

Validity of the findings

No comment

Additional comments

The authors have addressed my comments to my satisfaction.

Reviewer 2 ·

Basic reporting

I thank the authors for incorporating the requested changes. I think that the authors have addressed my concerns sufficiently.

However, I don’t entirely agree with their answer to the Point 10: “Why is the dynamics between 8 and 24h flat for some time in Figures 3 and S2 (and then jumps up)?”, to which the answer was (lines 273-277 in the revised manuscript): “Note that graphs for the number of virions contain flat sections. The reason for the first flat section is the latent period: all virus particles are removed by washing, and new particles have not been emitted yet. The following flat segments are the reflections of the first one, since virus production period of an infected cell is short due to resource exhaustion (see Appendix, Fig. S1), and new infected cells are in the latent period.”

While the first plateau after the dip is indeed due to latency in the virus production, the second plateau is not in my opinion related to latency in the virus production nor any resource exhaustion. First off, there is no resource exhaustion in the mathematical model. Second off, cytokines are in play, meaning, that cytokines make the healthy cells resistant for an extended period, meaning that the infected cells that can produce a second round of infectious particles are those which were initially infected. I might be wrong, but I think the authors should address this concern.

Lastly, I’d like the authors to explain why they chose to enforce correlation between the parameters beta and beta’ in the term for the infection of resistant cells beta’*V*U1=beta/k*V*U1. I think there is no reason for the infection rates of the resistant and susceptible cells to be correlated. They might be correlated, yes, but such a condition requires clarification of the reasoning behind it.

Experimental design

no comment

Validity of the findings

no comment

Additional comments

no comment

---

## Round 0.3 · accepted · Accept

Thank you for addressing the remaining concerns of the reviewer. I am glad to accept the revised manuscript.